# Synthetic Extracellular Matrices for 3D Culture of Schwann Cells, Hepatocytes, and HUVECs

**DOI:** 10.3390/bioengineering9090453

**Published:** 2022-09-08

**Authors:** Chiyuan Ma, Kaizheng Liu, Qin Li, Yue Xiong, Cuixiang Xu, Wenya Zhang, Changshun Ruan, Xin Li, Xiaohua Lei

**Affiliations:** 1Center for Energy Metabolism and Reproduction, Shenzhen Institutes of Advanced Technology, Chinese Academy of Sciences, Shenzhen 518055, China; 2Research Center for Human Tissue and Organs Degeneration, Shenzhen Institutes of Advanced Technology, Chinese Academy of Sciences, Shenzhen 518055, China; 3Shaanxi Provincial Key Laboratory of Infection and Immune Diseases, Shaanxi Provincial People’s Hospital, Xi’an 710068, China; 4Department of Stomatology, Shenzhen University General Hospital, Shenzhen University Clinical Medical Academy, 1098 Xuan Yuan Road, Nanshan District, Shenzhen 518055, China

**Keywords:** hydrogel, polyisocyanides, adhesive peptide, 3D culture

## Abstract

Synthetic hydrogels from polyisocyanides (PIC) are a type of novel thermoreversible biomaterials, which can covalently bind biomolecules such as adhesion peptides to provide a suitable extracellular matrix (ECM)-like microenvironment for different cells. Although we have demonstrated that PIC is suitable for three-dimensional (3D) culture of several cell types, it is unknown whether this hydrogel sustains the proliferation and passaging of cells originating from different germ layers. In the present study, we propose a 3D culture system for three representative cell sources: Schwann cells (ectoderm), hepatocytes (endoderm), and endothelial cells (mesoderm). Both Schwann cells and hepatocytes proliferated into multicellular spheroids and maintained their properties, regardless of the amount of cell-adhesive RGD motifs in long-term culture. Notably, Schwann cells grew into larger spheroids in RGD-free PIC than in PIC-RGD, while HL-7702 showed the opposite behavior. Endothelial cells (human umbilical vein endothelial cells, HUVECs) spread and formed an endothelial cell (EC) network only in PIC-RGD. Moreover, in a hepatocyte/HUVEC co-culture system, the characteristics of both cells were well kept for a long period in PIC-RGD. In all, our work highlights a simple ECM mimic that supports the growth and phenotype maintenance of cells from all germ layers in the long term. Our findings might contribute to research on biological development, organoid engineering, and in vitro drug screening.

## 1. Introduction

Cells reside in a highly complex 3D microenvironment [1] that is mainly composed of ECM in vivo [2]. In recent years, it has become increasingly clear that 2D cell culture models based on stiff planar substrates are not sufficient to simulate the complex processes that occur in soft 3D micromilieus in the human body [3,4]. Therefore, it is critical to establish an accurate 3D culture model to replicate the natural counterpart.

More and more reports show that the mechanical properties of ECM play an important role in the regulation of cell function [5]. Adhesive cells bind to their matrix through certain transmembrane proteins such as integrin and sense ECM mechanics through a large number of adhesion-related proteins and the downstream signaling pathways [6]. These physical signals, together with the classic biochemical signals, coordinate cell fates, including cell organization, proliferation, and migration [7,8].

Biomaterials have been widely used for 3D cell culture as artificial ECM in various forms, among which hydrogels are often selected due to their excellent biomimicry with natural matrices [9]. Hydrogels can be crudely divided into synthetic and natural hydrogels, and synthetic hydrogels are gaining the attention of many researchers due to their simplicity in composition and ease of single-factor control. Common synthetic hydrogels include polyvinyl alcohol (PVA) and polyethylene glycol (PEG) [10]. Once (covalently) crosslinked, these polymer networks provide mechanical support for various types of cells and are intrinsically biological inert: unlike natural hydrogels, where a plethora of endogenous factors can be found, unmodified synthetic hydrogels often do not possess any biological motif (conjugated and soluble) that interacts with cells [11]. Therefore, to better support cell growth, biofunctional peptide sequences (e.g., the classic cell adhesion peptide RGD) are often attached to the hydrogel backbone [12], and soluble factors such as growth factors can be added/immobilized into the system to further mimic the physiological condition [13]. To further simplify in vitro tissue culture, researchers are seeking 3D culture systems that are independent of biochemical cues, more convenient to use, and closer to the somatic environment to support cell proliferation and sustain cell properties.

Polyisocyanide (PIC) hydrogels, a novel type of synthetic hydrogels, have attracted an increasing amount of attention worldwide as they possess strongly biomimetic features in architecture and mechanics [14]. PIC forms gels at ultralow concentrations (typically 1 mg mL^−1^) above the gelation temperature (typical T_gel_ ≈ 20 °C) and liquefies once cooled down below T_gel_, which enables researchers to easily encapsulate and harvest cells. It has been previously reported that PIC hydrogels can support the growth of several cell types and mechanically modulate cell behavior [15]. Herein, we investigate the growth characteristics of cells from three germ layer sources (Schwann cells, hepatocytes, and ECs) and the maintenance of their phenotypes in PIC with(out) the RGD sequence. Results show that three types of cells exhibit different growth patterns in PIC and PIC-RGD. Schwann cells form larger spheroids in PIC without RGD, while hepatocytes vice versa. Endothelial cells only proliferate into networks in PIC-RGD. Moreover, we also test the hepatocyte/EC co-culture system and find out that these two cells can coexist and maintain their characteristics for a long period of culture even after multiple rounds of passaging. To summarize, we present a promising, simple, and stable platform for 3D culture and phenotype maintenance of somatic cells, which could potentially serve as a model for research on physiological and pathological development, as well as high-throughput drug screening in vitro.

## 2. Materials and Methods

### 2.1. Preparation and Characterization of PIC

PIC polymers were synthesized and biofunctionalized as previously reported [16]. Briefly, the isocyanide monomer was dissolved in anhydrous toluene and stirred. The ratio of azide-functionalized and total monomer was set to 1:30 and the ratio Ni^2+^ to total monomer 1:1000. The appropriate amounts of monomers and catalyst solution Ni(ClO_4_)_2_·6H_2_O (0.1 mg mL^–1^ in anhydrous toluene/absolute ethanol 9:1) were dissolved in toluene, and the final isocyanide concentration was adjusted to 50 mg mL^–1^. The polymer was precipitated in diisopropyl ether and collected by centrifugation after a 24 h reaction at room temperature. Thereafter, the polymer was dissolved in dichloromethane, precipitated for another two rounds, and air-dried into dark brown solids. The molecular weight of the polymer was determined by viscometry (dilute solutions in acetonitrile) using the empirical Mark–Houwink equation [η] = *KM*_v_*^a^*, where [η] is the experimentally determined intrinsic viscosity, *M*_v_ is the viscosity-determined molecular weight, and the Mark–Houwink constants *K* and *a* depend on polymer characteristics and solvent, temperature, etc. Parameters previously determined [17] for other polyisocyanides were used for calculation: *K* = 1.4 × 10^−9^ and *a* = 1.75.

The cell-adhesive GRGDS peptide was coupled with a DBCO-PEG4-NHS spacer and subsequently conjugated to the polymer through the SPAAC reaction, so that on average, 1% of the total monomers carried a peptide. PIC polymers were sterilized by UV and dissolved in culture medium overnight with a final concentration of 1 mg mL^−1^ (in cell-gel constructs), which corresponds to an RGD density of 0 and 28.5 μM for PIC and PIC-RGD, respectively. Rheology was performed on a stress-controlled rheometer (MCR302, Anton Paar, Graz, Austria) with a parallel plate geometry (diameter = 25 mm, gap = 400 μm) to measure the mechanical properties of the hydrogels. For the temperature ramp, the polymer solution was loaded onto the rheometer plate at T = 5 °C and heated to 37 °C (and back to 5 °C for the thermoreversibility check) at a rate of 1.0 °C min^−1^. Storage moduli were measured at a strain of *γ* = 0.02 and a frequency of *f* = 1.0 Hz.

For fluorescence labeling, a stock of 1 mg mL^−1^ TAMRA DBCO (Click Chemistry Tools) was mixed thoroughly with PIC solution on ice with a volume ratio of 1:2000. The mixture was incubated for 5 min for click reaction before thermo-gelation.

For cryoSEM imaging, the in situ gelation samples were mounted onto a preheated holder and quickly plunged into a freezing liquid nitrogen bath. A JEOL 6330 cryo-scanning electron microscope at an accelerating voltage of 3.0 kV was used.

### 2.2. Cell Culture

HL-7702 and HUVECs were purchased from BeNa culture collection (BNCC, Suzhou, China), and the rat Schwann cell (S16) was a kind gift from A.R. Du’s lab (Otwo Biotech, Shenzhen, China). All cells were cultured in DMEM (Gibco, 11995065, Grand Island, NE, USA) supplemented with 10% fetal bovine serum (Gibco, 10270-106, Grand Island, NE, USA) and 1% penicillin/streptomycin (final concentration of 100 IU mL^−1^ penicillin and 100 μg mL^−1^ streptomycin (Sigma-Aldrich, St. Louis, MO, USA).

### 2.3. Encapsulation and 3D Culture

Dry PIC polymers were sterilized by UV for 20 min and then dissolved in the medium for 24 h at 4 °C. Cells were harvested by trypsin treatment once they reached 100% confluence and were resuspended in fresh medium. Cell densities were determined by a NanoEnTek cell counter (NanoEntek, Inc. EVE-HT, Seoul, Korea) Cells were mixed with the polymer solution on ice with a predetermined ratio to achieve the required cell density (200,000 cells mL^−1^) and polymer concentration (1 mg mL^−1^). After mixing, the solutions were transferred to 96-well plates (Corning, Corning, NY, USA) or 8-well chambered cover slides (Lab-Tek, Beijing, China) and heated to 37 °C where gelation occurred. For all experiments, we transferred 100 μL cell-gel suspension to each well. After gelation, culture medium (37 °C) was gently added onto the samples. Then all samples were subject to standard cell culture conditions (37 °C, 5% CO_2_). For cell harvesting, the supernatant medium was replaced with an equal volume of cold PBS, and the whole plate was placed on ice. An amount of 5 mL cold PBS was added to further dilute the solution, followed by centrifugation at 1000 RPM for 5 min. The acquired cells could be passaged in the new hydrogels after trypsin digestion.

### 2.4. Bright Field Imaging and Cell Morphology Analysis

Bright field images of cells encapsulated in hydrogels were acquired on a Nikon TS-2R-FL inverted microscope. Quantitative analysis was performed by ImageJ (National Institutes of Health, v. 1.51J8, Bethesda, MD, USA). A paired sample t-test was used to determine the statistical significance.

### 2.5. Immunofluorescence (IF) Staining and Confocal Microscope Imaging

Gels with encapsulated cells were washed with PBS and then fixed with 4% paraformaldehyde in PBS for 40 min at room temperature. After fixation, the samples were permeabilized with 0.1% Triton X-100 in PBS for 60 min and blocked with 5% BSA in PBS for 60 min. They were then incubated with FITC-Phalloidin (50 IU mL^−1^ in 5% BSA/PBS, Sigma-Aldrich), CD31 (SouthernBiotech, 1625-01, Birmingham, AL, USA), ALB (Santa Cruz Biotechnology, sc-271605, Santa Cruz, CA, USA), and S100β (Abcam, ab52642, Cambridge, UK) overnight, respectively, later fluorescent secondary antibody and Hoechst33342 (5 mg mL^−1^ in PBS, Gibco, Grand Island, NE, USA) for 10 min. All procedures above were performed at 37 °C. A Lecia DMI8 confocal laser scanning microscope was used for fluorescence imaging. The temperature of the sample was kept at 37 °C by the heating element of the microscope. Images of three randomly chosen fields of vision (one-two spheroids per field) were captured.

### 2.6. Detection of Apoptosis (TUNEL Assay)

The terminal transferase dUTP nick end labeling (TUNEL) assay kit was purchased from Roche Diagnostic Corporation. For detection of apoptosis, TUNEL assay was performed following the manufacturer’s instructions. Briefly, the fixation process was performed as described above. After fixation, the samples were permeabilized with 0.1% Triton X-100 in PBS for 30 min at 37 °C. The TUNEL reaction was then performed using the manufacturer’s instructions. Hoechst33342 (5 mg mL^−1^ in PBS, Gibco) was used to visualize the nuclei. The state of cell clusters stained by the TUNEL was detected by a Lecia DMI8 confocal laser scanning microscope. Images of five to six randomly chosen fields of vision (one-two spheroids per field) were taken.

### 2.7. Statistical Analysis

All quantification data were presented as mean ± the standard error of the mean (SEM) with at least three independent replicates. Data were plotted and analyzed using GraphPad Prism (GraphPad Software, Inc. v.8.3.0, San Diego, CA, USA). Student’s *t*-test was used to determine the significance levels. Differences were regarded as significant once *p* < 0.05.

## 3. Results

### 3.1. Structure and Mechanical Properties of PIC Hydrogels

Two types of soft PIC matrices were prepared with or without cell-adhesive peptides (namely PIC and PIC-RGD), while the other parameters were kept constant, including the polymer backbone, polymer concentration, fibrous structure, and stiffness. Notably, PIC gels used in this study possess a storage modulus between 30 and 40 Pa, which is similar to the commonly used Matrigel matrices [5]. On top of the mechanics, the rapid thermoreversible sol-gel transition of PIC makes it highly convenient to harvest the encapsulated cells. Moreover, PIC gels show biomimetic nonlinear mechanics, also known as stress stiffening or strain stiffening, but it is not the focus of this study and will not be discussed in detail. See Figure 1 and Table 1 for the characterization of PIC polymers and hydrogels.

### 3.2. Growth and Characterization of Schwann Cell in PIC Hydrogels

We first investigate the functional supporting Schwann cell (S16) growth in PIC and PIC-RGD. Figure 2A shows that the dispersed single S16 cells formed obvious spheroids after 7 days of growing in both PIC and PIC-RGD hydrogels. On day 3, cells formed relatively uniform spheroids approximately 30–40 μm in diameter (Figure 2A,B). On day 7, spheroids grew relatively larger to 100 μm in diameter in PIC and 80 μm in PIC-RGD (Figure 2A). Interestingly, they did not spread in both hydrogels (Figure 2A). Further, the cell mass size was analyzed with bright field images by ImageJ. The maximum projected area was 2.2 × 10^4^ μm^2^ in PIC and 1.5 × 10^4^ μm^2^ in PIC-RGD after 7 days in culture, respectively (Figure 2B). To evaluate the rate of cell proliferation, we recovered all cell spheroids from each hydrogel by manipulating the temperature and digested the cell spheroids into single cells for counting. As shown in Figure 2C, S16 grew 3.29 times the initial cell number in PIC and 2.83 times in PIC-RGD after 3 days in culture, and that was approximately 4.22 times the original cell number in PIC and 4.36 times in PIC-RGD after 7 days in culture (Figure 2C). The cell proliferative capacity in RGD-containing gels was not significantly different from that in non-RGD-containing gels. The long-term stability of cell phenotypes is important to the development of 3D culture systems and for the facilitation of further organoid applications. To investigate whether PIC hydrogels could serve as a novel culture system for S16, S16 spheroids were routinely harvested and passaged in new PIC hydrogels of the same type (namely PIC to PIC, PIC-RGD to PIC-RGD). A stable proliferation rate could be still observed even after 10 rounds of harvesting and passaging (Figure 2D). To determine whether the cell grown in the gel still maintained the properties of S16 and explore their multicellular aggregated pattern, we performed immunofluorescence imaging of S16 marker molecules (central nervous specific protein, S100β) and cytoskeleton (F-actin) in the recovered cell spheroids. As shown in Figure 2E, an abundant S100β signal could be detected with clear F-actin boundaries between adjacent cells 7 days after the 10th round of passaging in both PIC and PIC-RGD. Image analysis shows that almost all cells are S100β positive in both PIC and PIC-RGD (Figure 2F). Then, to determine whether our 3D culture system causes apoptosis in cell spheroids, TUNEL staining, which measures DNA strand breaks, was performed. Results show that the TUNEL signal was hardly detected in all samples (Figure 2G,H), indicating good cell viability throughout spheroids.

### 3.3. Growth and Characterization of Hepatocytes in PIC Hydrogels

We further tested the 3D culture of hepatocytes in PIC and PIC-RGD hydrogel. As shown in Figure 3A, the first generation of HL-7702 cells encapsulated into PIC and PIC- RGD both formed obvious spheroids after 7 days. After 3 days in culture, cells generated relatively uniform spheroids approximately 40–50 μm in diameter (Figure 3A). After 7 days in culture, spheroids grew to relatively larger to 80 μm in diameter in PIC and 100 μm in PIC-RGD (Figure 3A). Additionally, they did not spread out in both hydrogels (Figure 3A). As clearly observed in Figure 3A, after 7 days of culture, HL-7702 grew into larger masses in PIC-RGD. To confirm such growth characteristics, cell mass size was analyzed with bright field images by ImageJ, too. The maximum projected area was 2 × 10^4^ μm^2^ and 3.8 × 10^4^ μm^2^ in PIC and PIC-RGD, respectively (Figure 3B). To evaluate the rate of cell proliferation, we also recovered all cell spheroids from each hydrogel and digested them into single cells to count the exact number. As shown in Figure 3C, HL-7702 grew 1.51 times the initial cell number in PIC and 1.94 times in PIC-RGD after 3 days in culture, and that was approximately 4.08 times the initial cell number in PIC and 4.01 times in PIC-RGD after 7 days in culture (Figure 3C). Further, we also performed 10 passages of HL-7702 in PIC and PIC-RGD, and stable proliferation of HL-7702 spheroids could be observed (Figure 3D), and stable expression of ALB after 7 days in the 10th generation (Figure 3E). Similarly, the cell grown in the gels still maintained the properties of HL-7702. The expression of the mature hepatocyte marker, albumin (ALB), was significant both in spheroids formed in PIC and PIC- RGD after 7 days in the culture of 10 generations and F-actin clearly separated each cell. (Figure 3E). Image analysis reveals that almost all cells are ALB positive, and no significant difference in percentage was observed between PIC and PIC-RGD (Figure 3F). Interestingly, it seems that HL-7702 formed more compact spheroids than S16 in both PIC and PIC-RGD. Moreover, the TUNEL signal was weak in all samples (Figure 3G), as confirmed by image quantification (Figure 3H).

### 3.4. Growth and Characterization of ECs in PIC Hydrogel

To investigate the growth of tissue-supporting EC in PIC hydrogels, we chose HUVECs, a common cell model for vascular EC research. As shown in Figure 4A, HUVECs encapsulated into PIC hydrogels with RGD showed proliferation and spread to form a vasoganglion-like network. To evaluate the rate of cell proliferation, we also count the accurate number of HUVECs after 3 and 7 days in culture. The cell number reached approximately 5.40 times the initial cell number in PIC-RGD after 7 days in culture. Meanwhile, in PIC, the cell number decreased, presumably due to the unsuitable environment for EC growth (Figure 4B). Note that only HUVECs grown in PIC-RGD were subcultured to PIC and PIC-RGD. The cells were still able to form networks and expressed CD31(the EC marker) after 5 passages in PIC-RGD, with a stable proliferation rate (Figure 4C,D). Further, the percentage of CD31 positive cells was calculated in three randomly chosen fields of vision. Almost all cells are CD31 positive (Figure 4E). Finally, the TUNEL staining performed in the EC network suggests a low apoptotic rate (Figure 4F), and the quantified result of the percentage of TUNEL negative cells was consistent with that (Figure 4G). Since HUVECs do not proliferate in PIC, the data do not apply to this group.

### 3.5. Co-Culture of ECs and Hepatocytes in PIC-RGD Enables Stable Cell Properties and EC Network Formation

Modeling interactions between human hepatocytes and ECs in vitro can help elucidate human-specific mechanisms underlying liver physiology/pathology and drug responses. Since ECs and hepatocytes sustain proliferation in PIC-RGD, we investigate if PIC-RGD also supports the co-culture of HUVECs and HL-7702. An equal amount of cell numbers is set for HUVECs and HL-7702, which means the initial total seeding density is 200,000 cells mL^−1^, and 100,000 cells mL^−1^ for each kind of cell, respectively. The formation of EC networks was visualized by bright-field microscopy and F-actin cytoskeletons of HUVECs were captured by fluorescence imaging (Figure 5A,B). As shown by the black arrow in Figure 5A, co-cultured HL-7702 and HUVECs formed a structure merging the morphologies of both cell types, which seemed to be spheroids with elongated cellular networks branching outwards. A stable expression of both cell markers (ALB and CD31) was confirmed after 3 rounds of passaging on day 7 (Figure 5B). Next, the percentage of ALB and CD31 positive cells was quantified in three randomly chosen fields of vision. In total, 19.58 ± 7.20% of cells are ALB positive, while 80.42 ± 4.16% are CD31 positive (Figure 5C). This is possibly due to the different rates of proliferation of these two cells in PIC-RGD. Additionally, there are no apparent TUNEL-positive cells in the co-cultured samples (Figure 5D), consistent with the quantification results (Figure 5E).

Together, these results suggest that both PIC and PIC-RGD are suitable for 3D culture of Schwann cells and hepatocytes, while PIC-RGD is suitable for ECs alone and co-culture with hepatocytes. The cell phenotypes can be sustained in long-term culture using the above-mentioned combinations.

## 4. Discussion

Three-dimensional culture has become an important research tool for studying the (patho)physiology of (ab)normal tissues, but there are still many aspects of the technology that need to be improved [18]. For instance, cell harvesting from 3D matrices often requires intense procedures to chemically destruct the hydrogels, which at the same induces cell damage. In this study, we developed a 3D cell culture platform based on thermoreversible gels that kept high and stable levels of phenotype marker expression of Schwann cells, hepatocytes, and HUVECs after several rounds of cell recycling. A schematic of the experimental procedure is depicted in Figure 6.

The use of biomaterials with combinations of hydrogels and bioactive peptides/natural ECM proteins has achieved desired effects on specific cell types [19,20,21]. In previous studies, the excellent properties of PIC have been demonstrated in the 3D culture of several cell types, including several cancer cells and mesenchymal stromal cells [22]. Here, we want to further verify whether PIC also performs well in the culture of other somatic cells. We came up with the idea of culturing representative cell types from three germ layers, which possess key functions in various systems of the human body, to expand the application of PIC in 3D tissue culture.

Schwann cells are a type of glial cells that develop from the ectoderm; hepatocytes are the major component of the liver. Previous studies have observed the spreading of both cell types in RGD-containing hydrogel matrices (alginate, fibrin, hyaluronic acid, PEG, etc.) [23,24,25,26]. However, in this study, neither Schwann cells nor hepatocytes spread out in PIC matrices; instead, both types of cells formed spheroids regardless of the presence of RGD. We believe that differences in mechanics, pore size, type, and density of biofunctionalization between PIC and other systems all play a role in the cellular outcome. Interestingly, the growth patterns of Schwann cells and hepatocytes also show differences in PIC hydrogels. S16 formed larger cell spheroids in plain PIC without RGD, while HL-7702 shows the opposite trend.

Previous reports confirm the interactions between these two cell types and the RGD peptide [25,26], which suggests that both cells can adhere to the matrix via integrin. However, the environmental complexity is enormous in the context of cell spheroids in a 3D fibrous ECM. At least three types of contact might simultaneously exist, namely cell–hydrogel, cell-cell, and cell–newly secreted ECM. All the above interactions may also be dominated by cell type and hydrogel properties. We hypothesize that different mechanobiological properties of the two cell types cause this phenomenon, which will be not discussed in detail here.

It is challenging to maintain phenotypes of hepatocytes after 1–2 weeks of in vitro culture [27]. Two-dimensional cultured hepatocytes show rapid declines in tissue-specific functions, such as cytochrome P-450 (CYP450) enzyme activities, insulin responsiveness, and expression of the master liver transcription factor hepatocyte nuclear factor 4α within hours to days [28]. As to 3D culture, studies have shown that hydrogels with RGD can well maintain the activity of hepatocytes for at least 9 days [26,29]. In our results, after 10 rounds of thermo-harvesting and passaging, hepatocytes still expressed the liver marker molecule ALB in PIC and PIC-RGD, which at least indicates that they can maintain cell characteristics without adding extra cytokines, regardless of the presence of cell adhesion sites.

Previous studies on the 3D culture of ECs have mostly used biologically derived matrices, such as Matrigel [30] or collagen/hyaluronic acid gels [27,31]. In this study, PIC-RGD as a synthetic counterpart of the above-mentioned natural gels also supports the growth and formation of the EC network of HUVECs. These results indicate that the RGD peptides conjugated to PIC gels function as anchorage sites for cells to bind to the matrix, subsequently spreading out and forming a network. Interestingly, unlike the two types of cells above, HUVEVs in plain PIC could not proliferate, whereas HUVECs in PIC-RGD proliferated approximately 5.4 times in 7 days and were well spread. This is in accordance with the other reports, where ECs were observed to spread more in an RGD-richer environment [32].

In vivo, vascular ECs cooperate with other somatic cells to sustain the growth and functioning of tissues. Therefore, researchers often co-culture vascular ECs with other substantial cells in vitro to achieve enhancement of cell function [33,34]. Taking the liver as an example, it has been explored to utilize ECs or nonparenchymal cells for transient enhancement of hepatocyte functions in co-cultures [35,36,37,38]. Unlike the previous studies that mostly involve rodent cells [39,40], in this study, we establish the co-culture system using human-derived cell lines for better modeling of human physiology. Results show good compatibility between hepatocytes and ECs in PIC-RGD. Both cells maintained their biological characteristics, including the expression of ALB (hepatocytes) and CD31 (ECs), after multiple rounds of harvesting and re-encapsulation. However, it needs to be further investigated whether this co-culture system mimics the functions of natural liver tissues.

To summarize, we employ thermoresponsive PIC matrices for the proliferation and phenotype maintenance of somatic cells. Cells are encapsulated in 3D, harvested by a simple step of cooling down, and reseeded for multiple rounds without influencing the viability and expression of tissue-specific markers, highlighting the biocompatibility, convenience, repeatability, and wide applicability of PIC for in vitro tissue culture. In the future, we aim to further modify PIC to fulfill the microenvironmental requirements of specific somatic cell types, such as stiffness, structure, and presentation of different biomolecules. We believe that this novel soft material holds great promise for research on developmental biology, physiology and pathology, and in vitro drug screening.

## Figures and Tables

**Figure 1 bioengineering-09-00453-f001:**
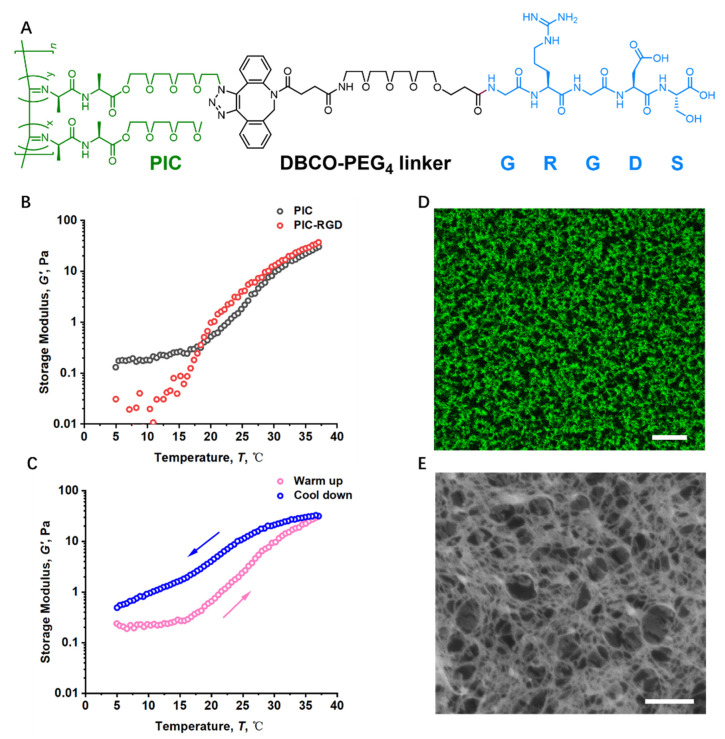
Structure and mechanical properties of PIC hydrogels. (**A**) Chemical structure of PIC-RGD; the GRGDS peptide is functionalized onto the side chain of PIC via a SPAAC reaction between azide and DBCO, 3.3% of the total monomers are appended with N_3_, and 1% of the total monomers is functionalized with GRGDS. (**B**,**C**) Thermoresponsive behavior of PIC. The temperature ramps show that PIC gelates upon heating with a transition temperature between 15–20 °C (**B**) and liquefies once the temperature declines below the gelation point (**C**). (**D**,**E**) The fibrous architecture revealed by fluorescence imaging ((**D**), scale bar =10 μm) and cryoSEM ((**E**), scale bar = 0.5 μm).

**Figure 2 bioengineering-09-00453-f002:**
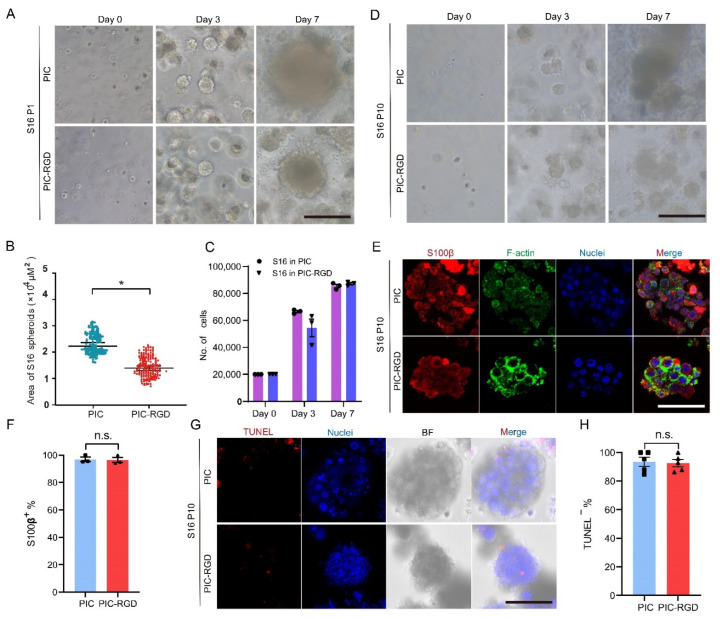
Characterization of rat Schwann cell (S16) in PIC hydrogels. (**A**) S16 cells were encapsulated in PIC and PIC-RGD and went through 7 days of culture (scale bar =100 μm). (**B**) The maximum projected area of cell spheroids under one field of vision on day 7. S16 shows larger spheroids formation in PIC than in PIC-RGD (* *p* < 0.05). (**C**) Number of S16 grown in PIC and PIC-RGD for 0, 3, and 7 days. The error bars represent the standard error of the mean of three experiments. (**D**) The growth characteristic of amplified cells is stable after 10 rounds of passaging in PIC and PIC-RGD. Note that S16 were subcultured always in the same type of matrices, either PIC or PIC-RGD (scale bar =100 μm). (**E**) Fluorescence staining of P10 S16 spheroids after 7 days in culture in PIC and PIC-RGD shows good maintenance of cellular characteristics. S100β: in red, Schwann cells maker; phalloidin: in green, stains F-actin and Hoechst 33342: in blue, stains cell nuclei (scale bar =100 μm). (**F**) Percentage of S100β positive cells per field of vision. (**G**)TUNEL assay (red) with Hoechst 33342 staining (blue, cell nuclei) and bright field image of P10 S16 after 7 days in culture in PIC and PIC-RGD show few apoptotic cells in spheroids (scale bar =100 μm). (**H**) Percentage of TUNEL negative cells per field of vision. For all samples, the PIC and PIC-RGD concentration c =1 mg mL^−1^, and the initial cell density is 200,000 cells mL^−1^. (n.s. means no significant differences).

**Figure 3 bioengineering-09-00453-f003:**
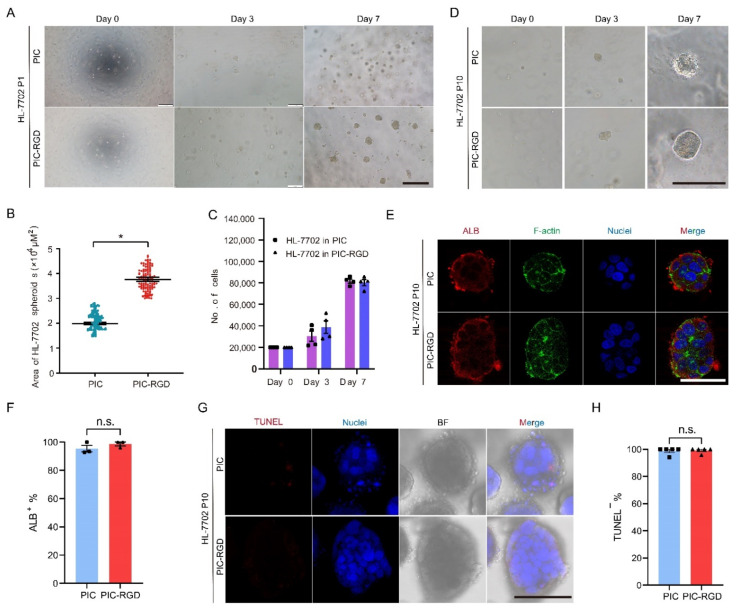
Characterization of HL-7702 in PIC hydrogels. (**A**) HL-7702 cells were encapsulated in PIC and PIC-RGD and went through 7 days of culture (scale bar =400 μm). (**B**) The maximum projected area of cell spheroids under one field of vision. HL-7702 formed larger spheroids in PIC-RGD than in PIC (* *p* < 0.05). (**C**) Number of HL-7702 grown in PIC and PIC-RGD for 0, 3, and 7 days. HL-7702 proliferated at a similar rate in PIC and PIC-RGD. The error bars represent the standard error of mean of four experiments. (**D**) The growth characteristic of amplified cells is stable after 10 generations of passaging in PIC and PIC-RGD. Note that HL-7702 were subcultured always in the same type of matrices, either PIC or PIC-RGD (scale bar =200 μm). (**E**) Fluorescence staining of P10 HL-7702 spheroids after 7 days of culture in PIC and PIC-RGD shows good maintenance of cellular characteristics. ALB: in red, hepatocyte maker; phalloidin: in green, stains F-actin and Hoechst33342: in blue, stains cell nuclei (scale bar =100 μm). (**F**) Percentage of ALB positive cells per field of vision. (**G**) TUNEL assay (red) with Hoechst33342 staining (blue, cell nuclei) and bright field image of P10 HL-7702 after 7 days in culture in PIC and PIC-RGD show few apoptotic cells in spheroids (scale bar =100 μm). (**H**) Percentage of TUNEL negative cells per field of vision. For all samples, the PIC and PIC-RGD concentration c = 1 mg mL^−1^ and the initial cell density is 200,000 cells mL^−1^. (n.s. means no significant differences).

**Figure 4 bioengineering-09-00453-f004:**
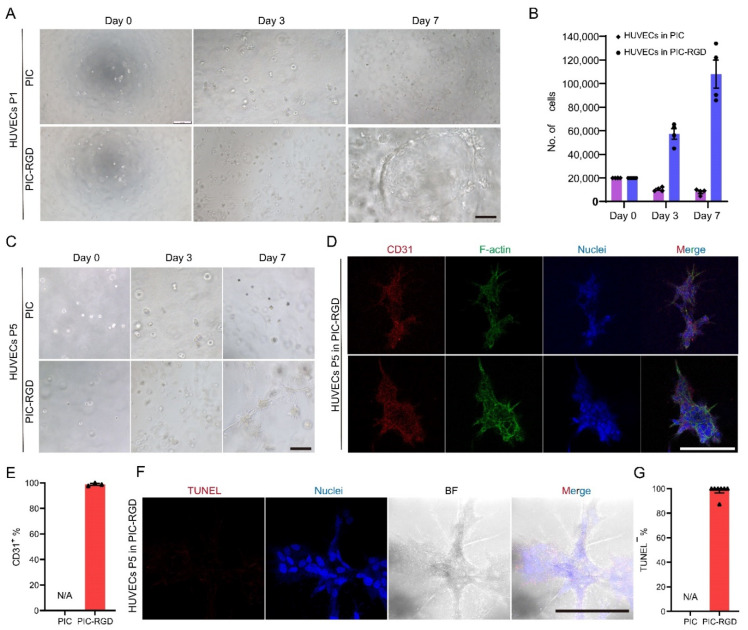
Characterization of HUVECs in PIC hydrogels. (**A**) HUVECs were encapsulated in PIC and PIC-RGD and went through 7 days of culture (scale bar = 200 μm). (**B**) Number of HUVECs grown in PIC and PIC-RGD for 0, 3, and 7 days. HUVECs did not grow in the PIC gel without RGD, but the proliferation of the PIC -RGD was significant. The error bars represent the standard error of mean of four experiments. (**C**) HUVECs grown in PIC-RGD were subcultured to PIC and PIC-RGD in each generation. The growth characteristic of HUVECs is stable after 5 rounds of passaging in PIC and PIC-RGD (scale bar = 200 μm). (**D**) Fluorescence staining of P5 HUVECS in PIC-RGD after 7 days of cultures shows good maintenance of phenotype characteristics. CD31: in red, EC maker; phalloidin: in green, stains F-actin; Hoechst33342: in blue, stains cell nuclei (scale bar = 200 μm). (**E**) Percentage of CD31 positive cells per field of vision. (**F**) TUNEL assay (red) with Hoechst33342 staining (blue, cell nuclei) and bright field image of P5 HUVECs after 7 days in culture in PIC-RGD show few apoptotic cells in spheroids (scale bar = 200 μm). (**G**) Percentage of TUNEL negative cells per field of vision. For all samples, the PIC and PIC-RGD concentration c = 1 mg mL^−1^ and the initial cell density is 200,000 cells mL^−1^. N/A represents not applicable data, since HUVECs do not proliferate in PIC.

**Figure 5 bioengineering-09-00453-f005:**
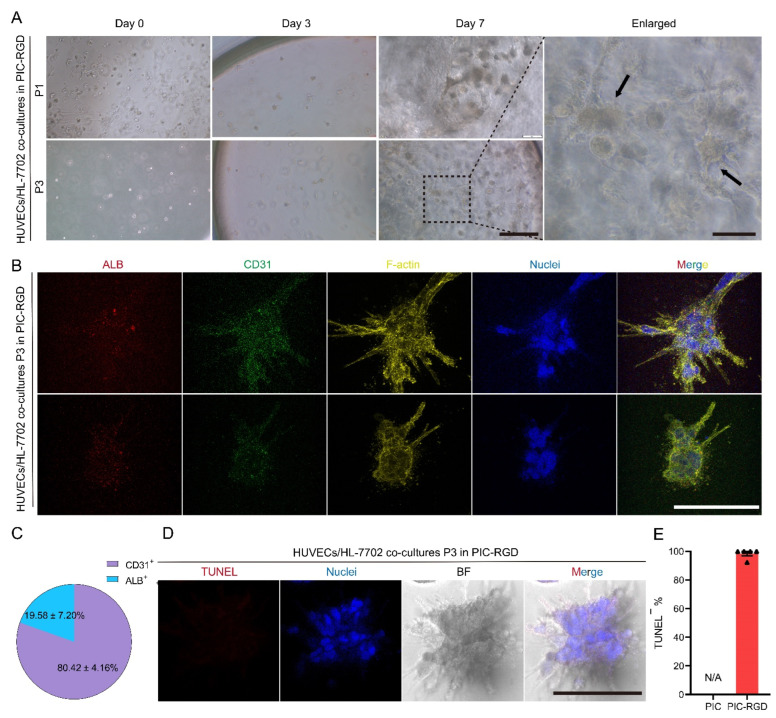
Co-culture of HUVECs with HL-7702 in PIC hydrogels. (**A**) HUVECs and HL-7702 cells were encapsulated in PIC-RGD and went through 7 days of culture and 3 rounds of passaging in PIC-RGD (scale bar = 200 μm). The enlarged image represents the dashed area in A, and black arrows show the representative structure of co-culture that differs from the culture of HUVECs alone (scale bar = 50 μm). (**B**) Fluorescence staining of co-culture in PIC-RGD shows good maintenance of cellular properties after 3 rounds of passaging in PIC-RGD. ALB: in red, HL-7702 maker; CD31: in green, EC maker; phalloidin: in yellow, stains F-actin, and Hoechst 33342: in blue, stains cell nuclei (scale bar = 50 μm). (**C**) Percentage of ALB and CD31 positive cells in four fields of vision. (**D**) TUNEL assay (red) with Hoechst 33342 staining (blue, cell nuclei) and bright field image of P3 co-culture of HUVECs and HL-7702 after 7 days in culture in PIC-RGD show few apoptotic cells in co-cultures (scale bar = 200 μm). (**E**) Percentage of TUNEL negative cells per field of vision. For all samples, the PIC-RGD concentration c = 1 mg mL^−1^ and the initial cell density is 200,000 cells mL^−1^ for mixed cells, and 100,000 cells mL^−1^ for each kind of cell, respectively. N/A represents not applicable data, since plain PIC does not support HUVECs/HL-7702 co-culture.

**Figure 6 bioengineering-09-00453-f006:**
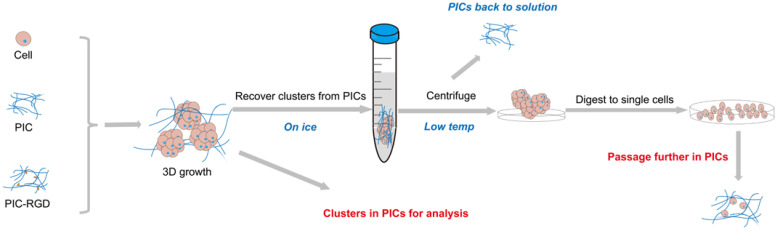
Schematic overview of the 3D culture process based on PIC hydrogels.

**Table 1 bioengineering-09-00453-t001:** PIC hydrogels used in the current study.

Hydrogel	Polymer	[Ni^2+^]:[M] Ratio ^a^	*M*_v_^b^ (kg mol^–1^)	*L*_C_^c^ (nm)	c (mg mL^–1^)	RGD content (μM)	*G’* at 37 °C (Pa)
PIC	Azide-appended PIC polymer	1:1000	363	144	1	0	30
PIC-RGD	GRGDS-functionalized PIC polymer	1:1000	363	144	1	28.5	37

^a^ Catalyst: monomer ratio for the polymerization reaction. The constant catalyst: monomer ratio ensures a constant average polymer contour length. ^b^ *M*_v_ = Viscosity-derived molecular weight of azide-appended polymers. ^c^ Average contour length based on *M*_v_.

## Data Availability

All data from this study are available from the authors upon request.

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
