# Peer review of "Synthetic Extracellular Matrices for 3D Culture of Schwann Cells, Hepatocytes, and HUVECs"

_bioengineering, 2022, doi:10.3390/bioengineering9090453_

Round 1
Reviewer 1 Report
Dear Authors,
I have reviewed with pleasure your paper and it seems to me a very good work.
I have no objection or comment to it. It is the first time that something similar happens to me.
Congratulations
Author Response
Bioengineering
Detailed response to comments from the editor and reviewers:
We appreciate the support and suggestions on our manuscript from the editor and the reviewers. Our manuscript has been carefully revised according to the comments provided. Changes in the manuscript have been highlighted in blue here in the replies. Moreover, we went through the manuscript and corrected a numer of detailed mistakes. Please, see below for our replies to each specific comment from the reviewers.
Reviewer 1
Dear Authors,
I have reviewed with pleasure your paper and it seems to me a very good work.
I have no objection or comment to it. It is the first time that something similar happens to me.
Congratulations
Reply: We appreciate the very positive feedback and kind support from the reviewer. We will then further address the remarks from the other reviewers.

Reviewer 2 Report
I have read with interest article entitled “Synthetic extracellular matrices for 3D culture od schwann cells, hepatocytes and HUVECs”. In order to improve manuscript quality some changes have to be performed:
1. Original source of S16 cell line should be provide.
2. UV light can be used for air or flat surface sterilization, better option for powder sterilization is ethylene oxide.
3. Figure 1C – in my opinion based on presented picture fibrous structure of material cannot be observed. For visualization of material structure scanning electron microscopy (SEM) should be used.
4. Analysis of cell markers and apoptosis should be improved. In presented form Authors showed only representative pictures from each tested cell line. In order to obtain numerical data from larger cell group relative fluorescence intensity from couple field of view should be performed or Authors should used other technique for measurement of cell markers like flow cytometry of western blot.
Author Response
Bioengineering
Detailed response to comments from the editor and reviewers:
We appreciate the support and suggestions on our manuscript from the editor and the reviewers. Our manuscript has been carefully revised according to the comments provided. Changes in the manuscript have been highlighted in blue here in the replies. Moreover, we went through the manuscript and corrected a numer of detailed mistakes. Please, see below for our replies to each specific comment from the reviewer. Please see the attachment.
Reviewer 2
I have read with interest article entitled “Synthetic extracellular matrices for 3D culture od schwann cells, hepatocytes and HUVECs”. In order to improve manuscript quality some changes have to be performed:
- Original source of S16 cell line should be provide.
Reply: We apologize for the missing of info, and have accordingly added the source of cells in section 2.2.
2.2. Cell Culture
HL-7702 and HUVECs were purchased from BeNa culture collection (BNCC), and rat Schwann cell (S16) was a kind gift from A.R. Du’s lab (Otwo Biotech). All cells cultured in DMEM (Gibco, 11995065) supplemented with 10% fetal bovine serum (Gibco, NA, 10270-106) and 1% penicillin/streptomycin (final concentration of 100 IU/mL penicillin and 100 μg/mL streptomycin, Sigma-Aldrich).
- UV light can be used for air or flat surface sterilization, better option for powder sterilization is ethylene oxide.
Reply: We thank the reviewer for raising this highly critical issue in future applications in biomedical research and clinics. The sterilization method of PIC hydrogel (aqueous state) has been investigated (Op’t Veld et al., 2020, https://doi.org/10.1089/ten.tec.2019.0305). But it remains systematically unexplored for dry-state PIC polymers. In practice, we have been using UV for polymer sterilization in the dry state, and so far contamination has rarely been observed. In the future, we will definitely follow the reviewer’s suggestion and test the effect of ethylene oxide treatment.
- Figure 1C – in my opinion based on presented picture fibrous structure of material cannot be observed. For visualization of material structure scanning electron microscopy (SEM) should be used.
Reply: We thank the reviewer for discussing the appropriate approach to depict the microscale and even nanoscale structure of PIC. The fibrous architecture has been confirmed by several reports using different techniques, including confocal microscopy (Vandaele et al., 2020, DOI: 10.1039/C9SM01828J), and cryoSEM (Yuan et al., 2017, https://doi.org/10.1021/acs.macromol.7b01832; Schoenmakers et al., 2018, https://doi.org/10.1038/s41467-018-04508-x). Here in this manuscript, for a comprehensive characterization of the fibrous structure of PIC, we have added a cryoSEM image in Figure 1 as suggested (Figure 1E).
Figure 1. Structure and mechanical properties of PIC hydrogels. (A) Chemical structure of PIC-RGD, the GRGDS peptide is functionalized onto the side chain of PIC via a SPAAC reaction between azide and DBCO, 3.3% of the total monomers are appended with N3, and 1% of the total monomers are functionalized with GRGDS. (B-C) Thermoresponsive behavior of PIC. The temperature ramps show that PIC gelates upon heating with a transition temperature between 15-20 °C (B), and liquefies once the temperature declines below the gelation point(C). (D-E) The fibrous architecture revealed by fluorescence imaging (D, scale bar =10 μm) and cryoSEM (E, scale bar = 0.5 μm).
- Analysis of cell markers and apoptosis should be improved. In presented form Authors showed only representative pictures from each tested cell line. In order to obtain numerical data from larger cell group relative fluorescence intensity from couple field of view should be performed or Authors should used other technique for measurement of cell markers like flow cytometry of western blot.
Reply: We thank the reviewer for this valuable technical remark. Indeed, quantitative analysis of cell markers and apoptosis is necessary for a comprehensive evaluation of the whole cell population. In this work, we focus on the question if 3D matrices based on PIC support the maintenance of cell phenotypes, therefore we have accordingly performed image analysis to quantify the rate of cells expressing cell markers. We highly agree with the reviewer that analysis on fluorescence intensity would be beneficial for more in-depth understanding when one compares different culture conditions, and this will be performed in our future work that aims to discuss the detailed parameters manipulating cell phenotypes and survival in 3D. The updated image analysis has been integrated into Figure 2-5. Here, we showcase the new Figure 2 as an example.
Figure 2. Characterization of rat Schwann cell (S16) in PIC hydrogels. (A) S16 cells were encapsulated in PIC and PIC-RGD and went through 7 days of culture (scale bar =100 μm). (B) The maximum projected area of cell spheroids under one field of vision at day 7. S16 shows larger spheroids formation in PIC than in PIC-RGD (* P<0.05). (C) Number of S16 grown in PIC and PIC-RGD for 0, 3, and 7 days. The error bars represent the standard error of mean of three experiments. (D) The growth characteristic of amplified cells is stable after 10 rounds of passageing in PIC and PIC-RGD. Note that S16 were subcultured always in the same type of matrices, either PIC or PIC-RGD (scale bar =100 μm). (E) Fluorescence staining of P10 S16 spheroids after 7 days in culture in PIC and PIC-RGD show good maintenance of cellular characteristics. S100β: in red, Schwann cells maker; phalloidin: in green, stains F-actin and Hoechst 33342: in blue, stains cell nuclei (scale bar =100 μm). (F) Percentage of S100β positive cells per field of vision. (G)TUNEL assay (red) with Hoechst 33342 staining (blue, cell nuclei) and bright field (BF) image of P10 S16 after 7 days in culture in PIC and PIC-RGD show that few apoptosis cells in spheroids (scale bar =100 μm). (H) Percentage of TUNEL negative cells per field of vision. For all samples, the PIC and PIC-RGD concentration c =1 mg mL−1, and the initial cell density is 200,000 cells mL−1.

Reviewer 3 Report
In this study Li and Lei and co-workers build up upon previous results which established hydrogels based on polyisocyanides (PIC) as thermo-reversible materials. Moreover, the authors demonstrated also previously that covalent functionalization of PIC hydrogels with peptide bioepitopes (e.g. RGD) resulted in novel 3D matrices, surrogates of the extracellular matrix, which could sustain growth of different cells - cancer cells and mesenchymal stromal cells. In this work the authors demonstrate that PIC and RGD-decorated PIC hydrogels are suitable surrogates of the extracellular matrix for the culture of cells representative of the ectoderm (Schwann cells), endoderm (hepatocytes) and mesoderm (endothelial cells).The authors demonstrate also that the RGD-PIC hydrogels are efficacious matrices for co-culture of hepatocytes and HUVECs.
This work is performed very competently, to high technical and scientific standards and is well presented and discussed.
Despite its incremental nature, this work represents a relevant contribuition towards the development of “universal” artificial surrogates of the extracellular matrix.
I believe that this study will appeal to many researchers in the biomaterials and tissue engineering research fields.
This work deserves publication after addressing some major issues:
I believe that it is not sufficient to refer to previous works when dealing with the chemical synthesis and characterization of the PIC and RGD-functionalized PIC hydrogels, especially when polymers (batch to batch variation, molecular weight distribution and dispersity….) are involved. I think that the authors must present as supporting information the procedures for the synthesis and the chemical characterization of the PIC polymers specifically used in the current study. For example, in table I the authors mention the “catalyst/monomer ratio”, but what is the nature of the catalyst?… The readers must be able to reproduce accurately the results described by the authors which will be determined to a great extention by the chemical properties of the PIC polymers.
Best regards
Author Response
Bioengineering
Detailed response to comments from the editor and reviewers:
We appreciate the support and suggestions on our manuscript from the editor and the reviewers. Our manuscript has been carefully revised according to the comments provided. Changes in the manuscript have been highlighted in blue here in the replies. Moreover, we went through the manuscript and corrected a numer of detailed mistakes. Please, see below for our replies to each specific comment from the reviewers. Please see the attachment.
Reviewer 3
In this study Li and Lei and co-workers build up upon previous results which established hydrogels based on polyisocyanides (PIC) as thermo-reversible materials. Moreover, the authors demonstrated also previously that covalent functionalization of PIC hydrogels with peptide bioepitopes (e.g. RGD) resulted in novel 3D matrices, surrogates of the extracellular matrix, which could sustain growth of different cells - cancer cells and mesenchymal stromal cells. In this work the authors demonstrate that PIC and RGD-decorated PIC hydrogels are suitable surrogates of the extracellular matrix for the culture of cells representative of the ectoderm (Schwann cells), endoderm (hepatocytes) and mesoderm (endothelial cells).The authors demonstrate also that the RGD-PIC hydrogels are efficacious matrices for co-culture of hepatocytes and HUVECs.
This work is performed very competently, to high technical and scientific standards and is well presented and discussed.
Despite its incremental nature, this work represents a relevant contribuition towards the development of “universal” artificial surrogates of the extracellular matrix.
I believe that this study will appeal to many researchers in the biomaterials and tissue engineering research fields.
Reply: We sincerely thank the reviewer for the precise and accurate summary of our manuscript, and the very enthusiastic and positive feedback.
This work deserves publication after addressing some major issues:
I believe that it is not sufficient to refer to previous works when dealing with the chemical synthesis and characterization of the PIC and RGD-functionalized PIC hydrogels, especially when polymers (batch to batch variation, molecular weight distribution and dispersity….) are involved. I think that the authors must present as supporting information the procedures for the synthesis and the chemical characterization of the PIC polymers specifically used in the current study. For example, in table I the authors mention the “catalyst/monomer ratio”, but what is the nature of the catalyst?… The readers must be able to reproduce accurately the results described by the authors which will be determined to a great extention by the chemical properties of the PIC polymers.
Best regards
Reply: We thank the reviewer for the professional and beneficial suggestions on material synthesis, and have accordingly added the detailed procedures in the manuscript. The updated section 2.1 reads as follows.
2.1. Preparation and characterization of PIC
PIC polymers were synthesized and biofunctionalized as previously reported [REF: Liu et al., 2020, doi: 10.1021/acsami.0c16208]. Briefly, the isocyanide monomer was dissolved in anhydrous toluene and stirred. The ratio of azide-functionalized and total monomer was set to 1:30 and the ratio Ni2+ to total monomer 1:1000. The appropriate amounts of monomers and catalyst solution Ni(ClO4)2·6H2O (0.1 mg ml–1 in anhydrous toluene/absolute ethanol 9:1) were dissolved in toluene, and the final isocyanide concentration was adjusted to 50 mg mL–1. The polymer was precipitated in diisopropyl ether and collected by centrifugation after a 24h reaction at room temperature. Thereafter, the polymer was dissolved in dichloromethane, precipitated for another two rounds, and air-dried into dark brown solids. The molecular weight of the polymer was determined by viscometry (dilute solutions in acetonitrile) using the empirical Mark–Houwink equation [η] = KMva, where [η] is the experimentally determined intrinsic viscosity, Mv is the viscosity-determined molecular weight, and the Mark–Houwink constants K and a depend on polymer characteristics and solvent, temperature, etc. Parameters previously determined [REF: van Beijnen et al., Macromolecules (1980), 13 (6), 1386-91] for other polyisocyanides were used for calculation: K = 1.4 × 10–9 and a = 1.75.
The cell-adhesive GRGDS peptide was coupled with a DBCO-PEG4-NHS spacer and subsequently conjugated to the polymer through the SPAAC reaction, so that on average, 1% of the total monomers carried a peptide. PIC polymers were sterilized by UV and dissolved in culture medium overnight with a final concentration of 1 mg mL-1 (in cell-gel constructs), which corresponds to a RGD density of 0 and 28.5 μM for PIC and PIC-RGD respectively.
Rheology was performed on a stress-controlled rheometer (MCR302, Anton Paar) with a parallel plate geometry (diameter = 25 mm, gap = 400 μm) to measure the mechanical properties of the hydrogels. For the temperature ramp, polymer solution was loaded onto the rheometer plate at T = 5 °C and heated to 37 °C at a rate of 1.0 °C min-1. Storage moduli were measured at a strain of γ = 0.02 and a frequency of f = 1.0 Hz.
For fluorescence labeling, a stock of 1 mg/ml TAMRA DBCO (Click Chemistry Tools) was mixed thoroughly with PIC solution on ice with a volume ratio of 1:2000. The mixture was incubated for 5 minutes for click reaction before thermo-gelation.
For cryoSEM imaging, the in situ gelation samples were mounted onto a preheated holder and quickly plunged into a freezing liquid nitrogen bath. A JEOL 6330 cryo-scanning electron microscope at an accelerating voltage of 3.0 kV was used.

Reviewer 4 Report
The manuscript described the use of a polyisocyanide (PIC) hydrogel as extracellular matrix for various kinds of cells. It illustrated how the RGD-peptide moiety affects the results of cell proliferation, morphology and functions after a prolong time of cultivation in the hydrogel. The authors claim their findings may provide a better 3-D
culture method to study cell/tissue’s functioning in vitro, which may be valuable in drug testing. The cells tested are: Schwann cells, hepatocytes and endothelial cells—cells belong to three different germ layers, respectively. In all, the work is scientifically sound, and the style of presentation is clear. However, the manuscript needs a major revision due to the inadequate description of methods and material characterization, and needs to strengthen their evidences to support the various claims made.
1. The synthesis and other molecular information about the PIC polymer was inadequately described. The article referred to (Ref. 15) did not have this information. Molecular weight and composition of PIC are necessary. Data from NMR, GPC or FTIR may be sufficient.
2. S-16 (Schwann cells) and HL-7702 (hepatocytes) all formed spheroids in the hydrogels with or without RGD, though with different sizes. There is no explanation for this behavior. Whether these cells make contact through the integrins to RGD should be checked. In the case for endothelial cells< there should be positive evidence for RGD-contact.
3. The claim that this synthetic ECM can keep the cells in their functioning state, the central claim of this paper, is only supported by immunostaining of a marker protein: ALB for hepatocytes , S100beta for Schwann cells and CD 31 for HUVECs. This claim needs to be strengthened by time-series data, some negative controls (e.g. 2D culture) and perhaps other more quantitative data, such as PCR or Western blots of other markers.
4. The advantage of PIC hydrogel as a cultivation ECM-like material is that it is thermal reversible, i.e. cells can be retrieved and re-seeded repeatedly, thus is able to prolong the functioning state of the particular cells. This point although mentioned in the text, and referred to Fig.6, but no quantified results were displayed.
All of the above points are very critical for the validity of this manuscript.
Author Response
Bioengineering
Detailed response to comments from the editor and reviewers:
We appreciate the support and suggestions on our manuscript from the editor and the reviewers. Our manuscript has been carefully revised according to the comments provided. Changes in the manuscript have been highlighted in blue here in the replies. Moreover, we went through the manuscript and corrected a numer of detailed mistakes. Please, see below for our replies to each specific comment from the reviewers. Please see the attachment.
Reviewer 4
The manuscript described the use of a polyisocyanide (PIC) hydrogel as extracellular matrix for various kinds of cells. It illustrated how the RGD-peptide moiety affects the results of cell proliferation, morphology and functions after a prolong time of cultivation in the hydrogel. The authors claim their findings may provide a better 3-D culture method to study cell/tissue’s functioning in vitro, which may be valuable in drug testing. The cells tested are: Schwann cells, hepatocytes and endothelial cells—cells belong to three different germ layers, respectively. In all, the work is scientifically sound, and the style of presentation is clear. However, the manuscript needs a major revision due to the inadequate description of methods and material characterization, and needs to strengthen their evidences to support the various claims made.
- The synthesis and other molecular information about the PIC polymer was inadequately described. The article referred to (Ref. 15) did not have this information. Molecular weight and composition of PIC are necessary. Data from NMR, GPC or FTIR may be sufficient.
Reply: We thank the reviewer for the professional feedback on material synthesis. We have updated section 2.1 and Table 1 to include more details on polymerization and molecular weight etc. The updated sections are listed below.
2.1. Preparation and characterization of PIC
PIC polymers were synthesized and biofunctionalized as previously reported [REF: Liu et al., 2020, doi: 10.1021/acsami.0c16208]. Briefly, the isocyanide monomer was dissolved in anhydrous toluene and stirred. The ratio of azide-functionalized and total monomer was set to 1:30 and the ratio Ni2+ to total monomer 1:1000. The appropriate amounts of monomers and catalyst solution Ni(ClO4)2·6H2O (0.1 mg ml–1 in anhydrous toluene/absolute ethanol 9:1) were dissolved in toluene, and the final isocyanide concentration was adjusted to 50 mg mL–1. The polymer was precipitated in diisopropyl ether and collected by centrifugation after a 24h reaction at room temperature. Thereafter, the polymer was dissolved in dichloromethane, precipitated for another two rounds, and air-dried into dark brown solids. The molecular weight of the polymer was determined by viscometry (dilute solutions in acetonitrile) using the empirical Mark–Houwink equation [η] = KMva, where [η] is the experimentally determined intrinsic viscosity, Mv is the viscosity-determined molecular weight, and the Mark–Houwink constants K and a depend on polymer characteristics and solvent, temperature, etc. Parameters previously determined [REF: van Beijnen et al., Macromolecules (1980), 13 (6), 1386-91] for other polyisocyanides were used for calculation: K = 1.4 × 10–9 and a = 1.75.
The cell-adhesive GRGDS peptide was coupled with a DBCO-PEG4-NHS spacer and subsequently conjugated to the polymer through the SPAAC reaction, so that on average, 1% of the total monomers carried a peptide. PIC polymers were sterilized by UV and dissolved in culture medium overnight with a final concentration of 1 mg mL-1 (in cell-gel constructs), which corresponds to a RGD density of 0 and 28.5 μM for PIC and PIC-RGD respectively.
Rheology was performed on a stress-controlled rheometer (MCR302, Anton Paar) with a parallel plate geometry (diameter = 25 mm, gap = 400 μm) to measure the mechanical properties of the hydrogels. For the temperature ramp, polymer solution was loaded onto the rheometer plate at T = 5 °C and heated to 37 °C at a rate of 1.0 °C min-1. Storage moduli were measured at a strain of γ = 0.02 and a frequency of f = 1.0 Hz.
For fluorescence labeling, a stock of 1 mg/ml TAMRA DBCO (Click Chemistry Tools) was mixed thoroughly with PIC solution on ice with a volume ratio of 1:2000. The mixture was incubated for 5 minutes for click reaction before thermo-gelation.
For cryoSEM imaging, the in situ gelation samples were mounted onto a preheated holder and quickly plunged into a freezing liquid nitrogen bath. A JEOL 6330 cryo-scanning electron microscope at an accelerating voltage of 3.0 kV was used.
Table 1. PIC hydrogels used in the current study.
Hydrogel |
Polymer |
[Ni2+]:[M] ratioa |
Mvb (kg mol–1)
|
LCc (nm)
|
c (mg mL–1) |
RGD content (μM) |
G′ at 37 °C (Pa) |
||
PIC |
Azide-appended PIC polymer |
1:1000 |
363 |
144 |
1 |
0 |
30 |
||
PIC-RGD |
GRGDS-functionalized PIC polymer |
1:1000 |
363 |
144 |
1 |
28.5 |
37 |
||
a Catalyst: monomer ratio for the polymerization reaction. The constant catalyst: monomer ratio ensures a constant average polymer contour length.
b Mv = Viscosity-derived molecular weight of azide-appended polymers.
c Average contour length based on Mv.
- S-16 (Schwann cells) and HL-7702 (hepatocytes) all formed spheroids in the hydrogels with or without RGD, though with different sizes. There is no explanation for this behavior. Whether these cells make contact through the integrins to RGD should be checked.
Reply: We thank the reviewer for raising this in-depth and very challenging biological question. Indeed, Schwann cells and hepatocytes both form spheroids, despite an opposite trend: the former grow into larger spheroids in PIC, and the latter larger in PIC-RGD. Previous reports are confirming the interactions between these two cell types and RGD peptide (Romano et al., 2015, http://dx.doi.org/10.1016/j.actbio.2014.10.008.; MacPherson et al., 2021, https://doi.org/10.1038/s41598-021-86016-5), which suggests that they do have the ability to adhere to the matrix via integrin. However, in the context of cell spheroids in a 3D fibrous ECM, there is an enormous environmental complexity. At least three types of contact might simultaneously exist, namely cell-hydrogel, cell-cell, and cell-newly secreted ECM. And all the above interactions may also be dominated by cell type and hydrogel property.
In this work, the two types of hydrogels used are both soft, fibrous matrices, and the major difference between PIC and PIC-RGD is the presence of cell adhesion sites. Interestingly, for both cell types, the proliferation between two matrices on day 7 was similar. Therefore, we propose that Schwan cells prefer to aggregate more in inert and confined matrices, while hepatocytes show a tendency to stay isolated in inert microenvironments and require cell-matrix interactions to promote cell-cell communication. Note that formation of spheroids might involve cell migration and merging of adjacent cell spheroids, which makes the situation even more complex. In all, we have to admit that it is very challenging to clarify this phenomenon due to the reasons mentioned above, and choose not to dig in at this direction. We kindly ask the reviewer for understanding. Nevertheless, we have added more relevant discussion in the text.
Previous reports are confirming the interactions between these two cell types and the RGD peptide [REF: Romano et al., 2015, http://dx.doi.org/10.1016/j.actbio.2014.10.008.; MacPherson et al., 2021, https://doi.org/10.1038/s41598-021-86016-5], which suggests that both cells can adhere to the matrix via integrin. However, in the context of cell spheroids in a 3D fibrous ECM, the environmental complexity is enormous. At least three types of contact might simultaneously exist, namely cell-hydrogel, cell-cell, and cell-newly secreted ECM. All the above interactions may also be dominated by cell type and hydrogel property. We hypothesize that different mechanobiological properties of the two cell types cause this phenomenon, which will be not discussed in detail here.
In the case for endothelial cells< there should be positive evidence for RGD-contact.
Reply: We thank the reviewer for commenting on the necessity of cell-matrix adhesion for endothelial cells. As shown in Figure 4, HUVECs do not spread or proliferate in inert PIC matrices; but spread and form networks, and proliferate in PIC-RGD. We believe this is strong evidence that endothelial cells require RGD contact.
- The claim that this synthetic ECM can keep the cells in their functioning state, the central claim of this paper, is only supported by immunostaining of a marker protein: ALB for hepatocytes , S100beta for Schwann cells and CD 31 for HUVECs. This claim needs to be strengthened by time-series data, some negative controls (e.g. 2D culture) and perhaps other more quantitative data, such as PCR or Western blots of other markers.
Reply: We thank the reviewer for this very professional comment on molecular biology. Indeed, quantitative analysis on the expression of maker proteins is necessary for a comprehensive evaluation of the whole cell population. In this work, we focus on the question if 3D matrices based on PIC support the maintenance of cell phenotypes, therefore we have accordingly performed image analysis to quantify the rate of cells expressing cell markers. We highly agree with the reviewer that time-series data including 2D negative controls would be beneficial for more in-depth understanding when one compares different culture conditions, and this will be performed in our future work that aims to discuss the detailed parameters manipulating cell phenotypes and survival in different microenvironments. The updated image analysis has been integrated into Figure 2-5. The new Figure 2 is showcased below as an example.
Figure 2. Characterization of rat Schwann cell (S16) in PIC hydrogels. (A) S16 cells were encapsulated in PIC and PIC-RGD and went through 7 days of culture (scale bar =100 μm). (B) The maximum projected area of cell spheroids under one field of vision on day 7. S16 shows larger spheroids formation in PIC than in PIC-RGD (* P<0.05). (C) Number of S16 grown in PIC and PIC-RGD for 0, 3, and 7 days. The error bars represent the standard error of mean of three experiments. (D) The growth characteristic of amplified cells is stable after 10 rounds of passage in PIC and PIC-RGD. Note that S16 were subcultured always in the same type of matrices, either PIC or PIC-RGD (scale bar =100 μm). (E) Fluorescence staining of P10 S16 spheroids after 7 days in culture in PIC and PIC-RGD show good maintenance of cellular characteristics. S100β: in red, Schwann cells maker; phalloidin: in green, stains F-actin and Hoechst 33342: in blue, stains cell nuclei (scale bar =100 μm). (F) Percentage of S100β positive cells per field of vision. (G)TUNEL assay (red) with Hoechst 33342 staining (blue, cell nuclei) and bright field (BF) image of P10 S16 after 7 days in culture in PIC and PIC-RGD show that few apoptosis cells in spheroids (scale bar =100 μm). (H) Percentage of TUNEL negative cells per field of vision. For all samples, the PIC and PIC-RGD concentration c =1 mg mL−1, and the initial cell density is 200,000 cells mL−1.
- The advantage of PIC hydrogel as a cultivation ECM-like material is that it is thermal reversible, i.e. cells can be retrieved and re-seeded repeatedly, thus is able to prolong the functioning state of the particular cells. This point although mentioned in the text, and referred to Fig.6, but no quantified results were displayed. All of the above points are very critical for the validity of this manuscript.
Reply: We thank the reviewer for mentioning the characteristic thermoreversibility of PIC gels. In fact, this phenomenon has been previously reported (Nature, 2013, https://doi.org/10.1038/nature11839; Nature Communications, 2018, https://doi.org/10.1038/s41467-018-04508-x). Nevertheless, to showcase this practical property for cell harvesting, we have accordingly added the quantification of thermoreversibility into Figure 1 (Figure 1C).
Figure 1. Structure and mechanical properties of PIC hydrogels. (A) Chemical structure of PIC-RGD, the GRGDS peptide is functionalized onto the side chain of PIC via a SPAAC reaction between azide and DBCO, 3.3% of the total monomers are appended with N3, and 1% of the total monomers are functionalized with GRGDS. (B-C) Thermoresponsive behavior of PIC. The temperature ramps show that PIC gelates upon heating with a transition temperature between 15-20 °C (B), and liquefies once the temperature declines below the gelation point(C). (D-E) The fibrous architecture revealed by fluorescence imaging (D, scale bar =10 μm) and cryoSEM (E, scale bar = 0.5 μm).

Round 2
Reviewer 2 Report
Authors respond almost to all my suggestions. In some cases Authors indicated that proposed analysis will be performed in the future, however Authors explanation in this regard was sufficient to me. I recommend this manuscript for publication in Bioengineering.
Reviewer 3 Report
Dear Authors, Dear Editor,
The authors fully addressed my suggestions. I believe that the manuscript can be published in its current form.
Best regards
Reviewer 4 Report
The revision of manuscript satisfies my comments, so it is publishable.